# Uprooting and Rerooting Higher-Order Graphical Models

**Mark Rowland**[*]
University of Cambridge
mr504@cam.ac.uk

**Adrian Weller**[*]
University of Cambridge and Alan Turing Institute
aw665@cam.ac.uk

## Abstract

The idea of uprooting and rerooting graphical models was introduced specifically for binary pairwise models by Weller [19] as a way to transform a model to any of a whole equivalence class of related models, such that inference on any one model yields inference results for all others. This is very helpful since inference, or relevant bounds, may be much easier to obtain or more accurate for some model in the class. Here we introduce methods to extend the approach to models with higher-order potentials and develop theoretical insights. In particular, we show that the triplet-consistent polytope TRI is unique in being 'universally rooted'. We demonstrate empirically that rerooting can significantly improve accuracy of methods of inference for higher-order models at negligible computational cost.

## 1 Introduction

Undirected graphical models with discrete variables are a central tool in machine learning. In this paper, we focus on three canonical tasks of inference: identifying a configuration with highest probability (termed maximum a posteriori or MAP inference), computing marginal probabilities of subsets of variables (marginal inference) and calculating the normalizing constant (partition function). All three tasks are typically computationally intractable, leading to much work to identify settings where exact polynomial-time methods apply, or to develop approximate algorithms that perform well.

Weller [19] introduced an elegant method which first *uproots* and then *reroots* a given model $M$ to any of a whole class of *rerooted* models $\{M_i\}$. The method relies on specific properties of binary pairwise models and makes use of an earlier construction which reduced MAP inference to the MAXCUT problem on the *suspension graph* $\nabla G$ (1; 2; 12; 19, see §3 for details). For many important inference tasks, the rerooted models are equivalent in the sense that results for any one model yield results for all others with negligible computational cost. This can be very helpful since various models in the class may present very different computational difficulties for inference.

Here we show how the idea may be generalized to apply to models with higher-order potentials over any number of variables. Such models have many important applications, for example in computer vision [6] or modeling protein interactions [5]. As for pairwise models, we again obtain significant benefits for inference. We also develop a deeper theoretical understanding and derive important new results. We highlight the following contributions:

- In §3-§4, we show how to achieve efficient uprooting and rerooting of binary graphical models with potentials of any order, while still allowing easy recovery of inference results.
- In §5, to simplify the subsequent analysis, we introduce *pure k-potentials* for any order $k$, which may be of independent interest. We show that there is essentially only one pure k-potential which we call the *even k-potential*, and that even $k$-potentials form a basis for all model potentials.
- In §6, we carefully analyze the effect of uprooting and rerooting on Sherali-Adams [11] relaxations $\mathbb{L}_r$ of the marginal polytope, for any order $r$. One surprising observation in §6.2 is that $\mathbb{L}_3$ (the

---

[*]Authors contributed equally.

triplet-consistent polytope or TRI) is unique in being *universally rooted*, in the sense that there is an affine score-preserving bijection between $\mathbb{L}_3$ for a model and $\mathbb{L}_3$ for each of its rerootings.

- In §7, our empirical results demonstrate that rerooting can significantly improve accuracy of inference in higher-order models. We introduce effective heuristics to choose a helpful rerooting.

Our observations have further implications for the many variational methods of marginal inference which optimize the sum of score and an entropy approximation over a Sherali-Adams polytope relaxation. These include the Bethe approximation (intimately related to belief propagation) and cluster extensions, tree-reweighted (TRW) approaches and logdet methods [12; 14; 16; 22; 24].

## 1.1 Background and discussion of theoretical contributions

Based on earlier connections in [2], [19] showed the remarkable result for pairwise models that the triplet-consistent polytope ($\mathbb{L}_3$ or TRI) is universally rooted (in the restricted sense defined in [19, Theorem 3]). This observation allowed straightforward strengthening of previously known results, for example: it was previously shown [23] that the LP relaxation on TRI (LP+TRI) is always tight for an 'almost-balanced' binary pairwise model, that is a model which can be rendered balanced by removing one variable [17]. Given [19, Theorem 3], this earlier result could immediately be significantly strengthened to [19, Theorem 4], which showed that LP+TRI is tight for a binary pairwise model provided only that *some rerooting exists* such that the rerooted model is almost balanced.

Following [19], it was natural to suspect that the universal rootedness property might hold for all (or at least some) $\mathbb{L}_r, r \geq 3$. This would have impact on work such as [10] which examines which signed minors must be forbidden to guarantee tightness of LP+$\mathbb{L}_4$. If $\mathbb{L}_4$ were universally rooted, then it would be possible to simplify significantly the analysis in [10].

Considering this issue led to our analysis of the mappings to symmetrized uprooted polytopes given in our Theorem 17. We believe this is the natural generalization of the lower order relationships of $\mathbb{L}_2$ and $\mathbb{L}_3$ to RMET and MET described in [2], though this direction was not clear initially.

With this formalism, together with the use of even potentials, we demonstrate our Theorems 20 and 21, showing that in fact TRI is unique in being universally rooted (and indeed in a stronger sense than given in [19]). We suggest that this result is surprising and may have further implications.

As a consequence, it is not possible to generate some quick theoretical wins by generalizing previous results as [19] did to derive their Theorem 4, but on the other hand we observe that rerooting may be helpful in practice for any approach using a Sherali-Adams relaxation other than $\mathbb{L}_3$. We verify the potential for significant benefits experimentally in §7.

## 2 Graphical models

A *discrete graphical model* $M[G(V, E), (\theta_{\mathcal{E}})_{\mathcal{E} \in E}]$ consists of: a *hypergraph* $G = (V, E)$, which has $n$ vertices $V = \{1, \ldots, n\}$ corresponding to the variables of the model, and *hyperedges* $E \subseteq \mathcal{P}(V)$, where $\mathcal{P}(V)$ is the powerset of $V$; together with *potential* functions $(\theta_{\mathcal{E}})_{\mathcal{E} \in E}$ over the hyperedges $\mathcal{E} \in E$. We consider binary random variables $(X_v)_{v \in V}$ with each $X_v \in \mathbb{X}_v = \{0, 1\}$. For a subset $U \subseteq V$, $x_U \in \{0, 1\}^U$ is a configuration of those variables $(X_v)_{v \in U}$. We write $\overline{x}_U$ for the *flipping* of $x_U$, defined by $\overline{x}_i = 1 - x_i \ \forall i \in U$. The joint probability mass function factors as follows, where the normalizing constant $Z = \sum_{x_V \in \{0,1\}^V} \exp(\text{score}(x_V))$ is the *partition function*:

$$p(x_V) = \frac{1}{Z} \exp\left(\text{score}(x_V)\right), \qquad \text{score}(x_V) = \sum_{\mathcal{E} \in E} \theta_{\mathcal{E}}(x_{\mathcal{E}}). \tag{1}$$

## 3 Uprooting and rerooting

Our goal is to map a model $M$ to any of a whole family of models $\{M_i\}$ in such a way that inference on any $M_i$ will allow us easily to recover inference results on the original model $M$. In this section we provide our mapping, then in §4 we explain how to recover inference results for $M$.

The uprooting mechanism used by Weller [19] first reparametrizes edge potentials to the form $\theta_{ij}(x_i, x_j) = -\frac{1}{2} W_{ij} \mathbb{1}[x_i \neq x_j]$, where $\mathbb{1}[\cdot]$ is the indicator function (a reparameterization modifies

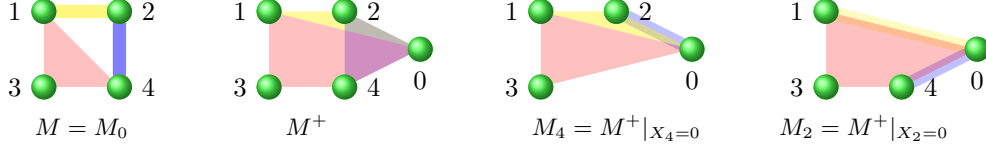

Figure 1: Left: The hypergraph $G$ of a graphical model $M$ over 4 variables, with potentials on the hyperedges $\{1,2\}$, $\{1,3,4\}$, and $\{2,4\}$. Center-left: The suspension hypergraph $\nabla G$ of the uprooted model $M^+$. Center-right: The hypergraph $\nabla G \setminus \{4\}$ of the rerooted model $M_4 = M^+|_{X_4=0}$, i.e. $M^+$ with $X_4$ clamped to 0. Right: The hypergraph $\nabla G \setminus \{2\}$ of the rerooted model $M_2 = M^+|_{X_2=0}$, i.e. $M^+$ with $X_2$ clamped to 0.

potential functions such that the complete score of each configuration is unchanged, see 15 for details). Next, singleton potentials are converted to edge potentials with this same form by connecting to an added variable $X_0$. This mechanism had been used previously to reduce MAP inference on $M$ to MAXCUT on the converted model [1; 12], and applies specifically only to binary pairwise models.

We introduce a generalized construction which applies to models with potentials of any order. We first *uproot* a model $M$ to a highly symmetric uprooted model $M^+$ where an extra variable $X_0$ is added, in such a way that the original model $M$ is exactly $M^+$ with $X_0$ *clamped* to the value 0. Since $X_0$ is clamped to retrieve $M$, we may write $M = M_0 := M^+|_{X_0=0}$. Alternatively, we can choose instead to clamp a different variable $X_i$ in $M^+$ which will lead to the *rerooted* model $M_i := M^+|_{X_i=0}$.

**Definition 1** (**Clamping**). For a graphical model $M[G = (V, E), (\theta_{\mathcal{E}})_{\mathcal{E} \in E}]$, and $i \in V$, the model $M|_{X_i=a}$ obtained by clamping the variable $X_i$ to the value $a \in \mathbb{X}_i$ is given by: the hypergraph $(V \setminus \{i\}, E_i)$, where $E_i = \{\mathcal{E} \setminus \{i\} | \mathcal{E} \in E\}$; and potentials which are unchanged for hyperedges which do not contain $i$, while if $i \in \mathcal{E}$ then $\theta_{\mathcal{E} \setminus \{i\}}(x_{\mathcal{E} \setminus \{i\}}) = \theta_{\mathcal{E}}(x_{\mathcal{E} \setminus \{i\}}, x_i = a)$.

**Definition 2** (**Uprooting, suspension hypergraph**). Given a model $M[G(V, E), (\theta_{\mathcal{E}})_{\mathcal{E} \in E}]$, the *uprooted* model $M^+$ adds a variable $X_0$, which is added to every hyperedge of the original model. $M^+$ has hypergraph $\nabla G$, with vertex set $V^+ = V \cup \{0\}$ and hyperedge set $E^+ = \{\mathcal{E}^+ = \mathcal{E} \cup \{0\} | \mathcal{E} \in E\}$. $\nabla G$ is the *suspension hypergraph* of $G$. $M^+$ has potential functions $(\theta^+_{\mathcal{E} \cup \{0\}})_{\mathcal{E} \in E}$ given by

$$\theta^+_{\mathcal{E} \cup \{0\}}(x_{\mathcal{E} \cup \{0\}}) = \begin{cases} \theta_{\mathcal{E}}(x_{\mathcal{E}}) & \text{if } x_0 = 0 \\ \theta_{\mathcal{E}}(\overline{x}_{\mathcal{E}}) & \text{if } x_0 = 1. \end{cases}$$

With this definition, all uprooted potentials are symmetric in that $\theta^+_{\mathcal{E}^+}(x_{\mathcal{E}^+}) = \theta^+_{\mathcal{E}^+}(\overline{x}_{\mathcal{E}^+}) \ \forall \mathcal{E}^+ \in E^+$.

**Definition 3** (**Rerooting**). From Definition 2, we see that given a model $M$, if we uproot to $M^+$ then clamp $X_0 = 0$, we recover the original model $M$. If instead in $M^+$ we clamp $X_i = 0$ for any $i = 1, \ldots, n$, then we obtain the *rerooted* model $M_i := M^+|_{X_i=0}$.

See Figure 1 and Table 1 for examples of uprooting and rerooting. We explore the question of how to choose a good variable for rerooting (i.e. how to choose a good variable to clamp in $M^+$) in §7.

## 4  Recovery of inference tasks

Here we demonstrate that the partition function, MAP score and configuration, and marginal distributions for a model $M$, can all be recovered from its uprooted model $M^+$ or any rerooted model $M_i$ $i \in V$, with negligible computational cost. We write $V_i = \{0, 1, \ldots, n\} \setminus \{i\}$ for the variable set of rerooted model $M_i$; $\text{score}_i(x_{V_i})$ for the score of $x_{V_i}$ in $M_i$; and $p_i$ for the probability distribution for $M_i$. We use superscript $+$ to indicate the uprooted model. For example, the probability distribution for $M^+$ is given by $p^+(x_{V^+}) = \frac{1}{Z^+} \exp\left(\sum_{\mathcal{E} \in E^+} \theta_{\mathcal{E}}(x_{\mathcal{E}})\right)$. From the definitions of §3, we obtain the following key lemma, which is critical to enable recovery of inference results.

**Lemma 4** (**Score-preserving map**). *Each configuration $x_V$ of $M$ maps to 2 configurations of the uprooted $M^+$ with the same score, i.e. from $M, x_V \to$ in $M^+$, both of $(x_0 = 0, x_V)$ and $(x_0 = 1, \overline{x}_V)$ with $\text{score}(x_V) = \text{score}^+(x_0 = 0, x_V) = \text{score}^+(x_0 = 1, \overline{x}_V)$. For any $i \in V^+$, exactly one of the two uprooted configurations has $x_i = 0$, and just this one will be selected in $M_i$. Hence, there is a score-preserving bijection between configurations of $M$ and those of $M_i$:*

$$\text{For any } i \in V^+: \quad \text{in } M, x_V \ \leftrightarrow \ \text{in } M_i, \begin{cases} (x_0 = 0, x_{V \setminus \{i\}}) & \text{if } x_i = 0 \\ (x_0 = 1, \overline{x}_{V \setminus \{i\}}) & \text{if } x_i = 1. \end{cases} \tag{2}$$

| $M$ config | | | $M^+$ configuration | | | | $M_4$ config | | |
| --- | --- | --- | --- | --- | --- | --- | --- | --- | --- |
| $x_1$ | $x_3$ | $x_4$ | $x_0$ | $x_1$ | $x_3$ | $x_4$ | $x_0$ | $x_1$ | $x_3$ |
| 0 | 0 | 0 | 0 | 0 | 0 | 0 | 0 | 0 | 0 |
| 0 | 0 | 1 | 0 | 0 | 0 | 1 | | | |
| 0 | 1 | 0 | 0 | 0 | 1 | 0 | 0 | 0 | 1 |
| 0 | 1 | 1 | 0 | 0 | 1 | 1 | | | |
| 1 | 0 | 0 | 0 | 1 | 0 | 0 | 0 | 1 | 0 |
| 1 | 0 | 1 | 0 | 1 | 0 | 1 | | | |
| 1 | 1 | 0 | 0 | 1 | 1 | 0 | 0 | 1 | 1 |
| 1 | 1 | 1 | 0 | 1 | 1 | 1 | | | |
| | | | 1 | 0 | 0 | 0 | 1 | 0 | 0 |
| | | | 1 | 0 | 0 | 1 | | | |
| | | | 1 | 0 | 1 | 0 | 1 | 0 | 1 |
| | | | 1 | 0 | 1 | 1 | | | |
| | | | 1 | 1 | 0 | 0 | 1 | 1 | 0 |
| | | | 1 | 1 | 0 | 1 | | | |
| | | | 1 | 1 | 1 | 0 | 1 | 1 | 1 |
| | | | 1 | 1 | 1 | 1 | | | |

Table 1: An illustration of how scores of potential $\theta_{134}$ on hyperedge $\{1, 3, 4\}$ in an original model $M$ map to potential $\theta_{0134}$ in $M^+$ and then to $\theta_{013}$ in $M_4$. See Figure 1 for the hypergraphs. Each color indicates a value of $\theta_{134}(x_1, x_3, x_4)$ for a different configuration $(x_1, x_3, x_4)$. Note that $M^+$ has 2 rows of each color, while after rerooting to $M_4$, we again have exactly one row of each color. The 1-1 score preserving map between configurations of $M$ and any $M_i$ is critical to enable recovery of inference results; see Lemma 4.

Table 1 illustrates this perhaps surprising result, from which the next two propositions follow.

**Proposition 5** (**Recovering the partition function**). Given a model $M[G(V, E), (\theta_{\mathcal{E}})_{\mathcal{E} \in E}]$ with partition function $Z$ as in (1), the partition function $Z^+$ of the uprooted model $M^+$ is twice $Z$, and the partition function of each rerooted model $M_i$ is exactly $Z$, for any $i \in V$.

**Proposition 6** (**Recovering a MAP configuration**). From $M^+$: $x_V$ is an arg max for $p$ iff $(x_0 = 0, x_V)$ is an arg max for $p^+$ iff $(x_0 = 1, \overline{x}_V)$ is an arg max for $p^+$. From a rerooted model $M_i$: $(x_{V \setminus \{i\}}, x_i = 0)$ is an arg max for $p$ iff $(x_0 = 0, x_{V \setminus \{i\}})$ is an arg max for $p_i$; $(x_{V \setminus \{i\}}, x_i = 1)$ is an arg max for $p$ iff $(x_0 = 1, \overline{x}_{V \setminus \{i\}})$ is an arg max for $p_i$.

We can recover marginals as shown in the following proposition, proof in the Appendix §9.1.

**Proposition 7** (**Recovering marginals**). For a subset $\emptyset \neq U \subseteq V$, we can recover from $M^+$:
$p(x_U) = p^+(x_0 = 0, x_U) + p^+(x_0 = 1, \overline{x}_U) \qquad = 2p^+(x_0 = 0, x_U) = 2p^+(x_0 = 1, \overline{x}_U)$.
To recover from a rerooted $M_i$: (i) For any $i \in V \setminus U$, $p(x_U) = p_i(x_0 = 0, x_U) + p_i(x_0 = 1, \overline{x}_U)$.

(ii) For any $i \in U$, $p(x_U) = \begin{cases} p_i(x_0 = 0, x_{U \setminus \{i\}}) & x_i = 0 \\ p_i(x_0 = 1, \overline{x}_{U \setminus \{i\}}) & x_i = 1. \end{cases}$

In §6, we provide a careful analysis of the impact of uprooting and rerooting on the Sherali-Adams hierarchy of relaxations of the marginal polytope [11]. We first introduce a way to parametrize potentials which will be particularly useful, and which may be of independent interest.

# 5 Pure $k$-potentials

We introduce the notion of *pure $k$-potentials*. These allow the specification of interactions which act 'purely' over a set of variables of a given size $k$, without influencing the distribution of any subsets. We show that in fact, there is essentially only one pure $k$-potential. Further, we show that one can express any $\theta_{\mathcal{E}}$ potential in terms of pure potentials over $\mathcal{E}$ and subsets of $\mathcal{E}$, and that pure potentials have appealing properties when uprooted and rerooted which help our subsequent analysis.

We say that a potential is a *$k$-potential* if $k$ is the smallest number such that the score of the potential may be determined by considering the configuration of $k$ variables. Usually a potential $\theta_{\mathcal{E}}$ is a $k$-potential with $k = |\mathcal{E}|$. For example, typically a singleton potential is a 1-potential, and an edge potential is a 2-potential. However, note that $k < |\mathcal{E}|$ is possible if one or more variables in $\mathcal{E}$ are not needed to establish the score (a simple example is $\theta_{12}(x_1, x_2) = x_1$, which clearly is a 1-potential).

In general, a $k$-potential will affect the marginal distributions of all subsets of the $k$ variables. For example, one popular form of 2-potential is $\theta_{ij}(x_i, x_j) = W_{ij} x_i x_j$, which tends to pull $X_i$ and $X_j$ toward the same value, but also tends to increase each of $p(X_i = 1)$ and $p(X_j = 1)$. For pairwise models, a different reparameterization of potentials instead writes the score as

$$\text{score}(x_V) = \sum_{i \in V} \theta_i x_i + \frac{1}{2} \sum_{(i,j) \in E} W_{ij} \mathbb{1}[x_i = x_j]. \tag{3}$$

Expression (3) has the desirable feature that the $\theta_{ij}(x_i, x_j) = \frac{1}{2} W_{ij} \mathbb{1}[x_i = x_j]$ edge potentials affect only the pairwise marginals, without disturbing singleton marginals. This motivates the following definition.

**Definition 8.** Let $k \geq 2$, and let $U$ be a set of size $k$. We say that a $k$-potential $\theta_U : \{0,1\}^U \to \mathbb{R}$ is a *pure $k$-potential* if the distribution induced by the potential, $p(x_U) \propto \exp(\theta_U(x_U))$, has the property that for any $\emptyset \neq W \subsetneq U$, the marginal distribution $p(x_W)$ is uniform.

We shall see in Proposition 10 that a pure $k$-potential must essentially be an *even $k$-potential*.

**Definition 9.** Let $k \in \mathbb{N}$, and $|U| = k$. An *even $k$-potential* is a $k$-potential $\theta_U : \{0,1\}^U \to \mathbb{R}$ of the form $\theta_U(x_U) = a \mathbb{1}[\, |\{i \in U | x_i = 1\}|$ is even$]$, for some $a \in \mathbb{R}$ which is its *coefficient*. In words, $\theta_U(x_U)$ takes value $a$ if $x_U$ has an even number of 1s, else it takes value 0.

As an example, the 2-potential $\theta_{ij}(x_i, x_j) = \frac{1}{2} W_{ij} \mathbb{1}[x_i = x_j]$ in (3) is an even 2-potential with $U = \{i,j\}$ and coefficient $W_{ij}/2$. The next two propositions are proved in the Appendix §9.2.

**Proposition 10** (All pure potentials are essentially even potentials)**.** Let $k \geq 2$, and $|U| = k$. If $\theta_U : \{0,1\}^U \to \mathbb{R}$ is a pure $k$-potential then $\theta_U$ must be an affine function of the even $k$-potential, i.e. $\exists\, a, b \in \mathbb{R}$ s.t. $\theta_U(x_U) = a \mathbb{1}[\, |\{i \in U | x_i = 1\}|$ is even$] + b$.

**Proposition 11** (Even $k$-potentials form a basis)**.** For a finite set $U$, the set of even $k$-potentials $\left( \mathbb{1}[\, |\{i \in W | X_i = 1\}|$ is even$] \right)_{W \subseteq U}$, indexed by subsets $W \subseteq U$, forms a basis for the vector space of all potential functions $\theta : \{0,1\}^U \to \mathbb{R}$.

Any constant in a potential will be absorbed into the partition function $Z$ and does not affect the probability distribution, see (1). An even 2-potential with positive coefficient, e.g. as in (3) if $W_{ij} > 0$, is *supermodular*. Models with only supermodular potentials (equivalently, submodular cost functions) typically admit easier inference [3; 7]; if such a model is binary pairwise then it is called *attractive*. However, for $k > 2$, even $k$-potentials $\theta_{\mathcal{E}}$ are neither supermodular nor submodular. Yet if $k$ is an even number, observe that $\theta_{\mathcal{E}}(x_{\mathcal{E}}) = \theta_{\mathcal{E}}(\overline{x}_{\mathcal{E}})$. We discuss this further in Appendix §10.4.

When a $k$-potential is uprooted, in general it may become a $(k+1)$-potential (recall Definition 2). The following property of even $k$-potentials is helpful for our analysis in §6, and is easily checked.

**Lemma 12** (Uprooting an even $k$-potential)**.** *When an even $k$-potential $\theta_{\mathcal{E}}$ with $|\mathcal{E}| = k$ is uprooted: if $k$ is an even number, then the uprooted potential is exactly the same even $k$-potential; if $k$ is odd, then we obtain the even $(k+1)$-potential over $\mathcal{E} \cup \{0\}$ with the same coefficient as the original $\theta_{\mathcal{E}}$.*

## 6 Marginal polytope and Sherali-Adams relaxations

We saw in Lemma 4 that there is a score-preserving 1-2 mapping from configurations of $M$ to those of $M^+$, and a bijection between configurations of $M$ and any $M_i$. Here we examine the extent to which these score-preserving mappings extend to (pseudo-)marginal probability distributions over variables by considering the Sherali-Adams relaxations [11] of the respective marginal polytopes. These relaxations feature prominently in many approaches for MAP and marginal inference.

For $U \subseteq V$, we write $\mu_U$ for a probability distribution in $\mathscr{P}(\{0,1\}^U)$, the set of all probability distributions on $\{0,1\}^U$. Bold $\boldsymbol{\mu}$ will represent a collection of measures over various subsets of variables. Given (1), to compute an expected score, we need $(\mu_{\mathcal{E}})_{\mathcal{E} \in E}$. This motivates the following.

**Definition 13.** The *marginal polytope* $\mathbb{M}(G(V, E)) = \{ (\mu_{\mathcal{E}})_{\mathcal{E} \in E} | \exists \mu_V$ s.t. $\mu_{V \downarrow \mathcal{E}} = \mu_{\mathcal{E}} \; \forall \mathcal{E} \in E \}$, where for $U_1 \subseteq U_2 \subseteq V$, $\mu_{U_2 \downarrow U_1}$ denotes the marginalization of $\mu_{U_2} \in \mathscr{P}(\{0,1\}^{U_2})$ onto $\{0,1\}^{U_1}$.

$\mathbb{M}(G)$ consists of marginal distributions for every hyperedge $\mathcal{E} \in E$ such that all the marginals are consistent with a global distribution over all variables $V$. Methods of variational inference typically

optimize either the score (for MAP inference) or the score plus an entropy term (for marginal inference) over a relaxation of the marginal polytope [15]. This is because $\mathbb{M}(G)$ is computationally intractable, with an exponential number of facets [2]. Relaxations from the Sherali-Adams hierarchy [11] are often used, requiring consistency only over smaller clusters of variables.

**Definition 14.** Given an integer $r \geq 2$, if a hypergraph $G(V, E)$ satisfies $\max_{\mathcal{E} \in E} |\mathcal{E}| \leq r \leq |V|$, then we say that $G$ is *r-admissible*, and define the *Sherali-Adams polytope of order $r$* on $G$ by

$$\mathbb{L}_r(G) = \left\{ (\mu_{\mathcal{E}})_{\mathcal{E} \in E} \,\middle|\, \exists (\mu_U)_{\substack{U \subseteq V \\ |U| = r}} \text{ locally consistent, s.t. } \mu_{U \downarrow \mathcal{E}} = \mu_{\mathcal{E}} \quad \forall\, \mathcal{E} \subseteq U \subseteq V, \ |U| = r \right\},$$

where a collection of measures $(\mu_A)_{A \in I}$ (for some set $I$ of subsets of $V$) is *locally consistent*, or l.c., if for any $A_1, A_2 \in I$, we have $\mu_{A_1 \downarrow A_1 \cap A_2} = \mu_{A_2 \downarrow A_1 \cap A_2}$. Each element of $\mathbb{L}_r(G)$ is a set of locally consistent probability measures over the hyperedges. Note that $\mathbb{M}(G) \subseteq \mathbb{L}_r(G) \subseteq \mathbb{L}_{r-1}(G)$. The pairwise relaxation $\mathbb{L}_2(G)$ is commonly used but higher-order relaxations achieve greater accuracy, have received significant attention [10; 13; 18; 22; 23], and are required for higher-order potentials.

## 6.1 The impact of uprooting and rerooting on Sherali-Adams polytopes

We introduce two variants of the Sherali-Adams polytopes which will be helpful in analyzing uprooted models. For a measure $\mu_U \in \mathscr{P}(\{0,1\}^U)$, we define the *flipped measure* $\overline{\mu}_U$ as $\overline{\mu}_U(x_U) = \mu_U(\overline{x}_U) \ \forall x_U \in \{0,1\}^U$. A measure $\mu_U$ is *flipping-invariant* if $\mu_U = \overline{\mu}_U$.

**Definition 15.** The symmetrized Sherali-Adams polytopes for an uprooted hypergraph $\nabla G(V^+, E^+)$ (as given in Definition 2), is:

$$\widetilde{\mathbb{L}}_r(\nabla G) = \left\{ (\mu_{\mathcal{E}})_{\mathcal{E} \in E^+} \in \mathbb{L}_r(\nabla G) \,\middle|\, \mu_{\mathcal{E}} = \overline{\mu}_{\mathcal{E}} \ \forall \mathcal{E} \in E^+ \right\}.$$

**Definition 16.** For any $i \in V^+$, and any integer $r \geq 2$ such that $\max_{\mathcal{E} \in E^+} |\mathcal{E}| \leq r \leq |V^+|$, we define the symmetrized Sherali-Adams polytope of order $r$ *uprooted at $i$* to be

$$\widetilde{\mathbb{L}}_r^i(\nabla G) = \left\{ (\mu_{\mathcal{E}})_{\mathcal{E} \in E^+} \,\middle|\, \exists (\mu_U)_{\substack{i \in U \subseteq V^+ \\ |U| = r}} \text{ l.c., s.t. } \begin{array}{l} \mu_{U \downarrow \mathcal{E}} = \mu_{\mathcal{E}} \quad \forall\, \mathcal{E} \subseteq U \subseteq V, \ |U| = r, i \in U \\ \mu_U = \overline{\mu}_U \quad \forall U \subseteq V, |U| = r, i \in U \end{array} \right\}.$$

Thus, for each collection of measures over hyperedges in $\widetilde{\mathbb{L}}_r^i(\nabla G)$, there exist corresponding flipping-invariant, locally consistent measures on sets of size $r$ which contain $i$ (and their subsets). Note that for any hypergraph $G(V, E)$ and any $i \in V^+$, we have $\widetilde{\mathbb{L}}_{r+1}(\nabla G) \subseteq \widetilde{\mathbb{L}}_{r+1}^i(\nabla G) \subseteq \widetilde{\mathbb{L}}_r(\nabla G)$. We next extend the correspondence of Lemma 4 to collections of locally-consistent probability distributions on the hyperedges of $G$, see the Appendix §9.3 for proof.

**Theorem 17.** *For a hypergraph $G(V, E)$, and integer $r$ such that $\max_{\mathcal{E} \in E} |\mathcal{E}| \leq r \leq |V|$, there is an affine score-preserving bijection*

$$\mathbb{L}_r(G) \overset{\textbf{Uproot}}{\underset{\textbf{RootAt0}}{\rightleftarrows}} \widetilde{\mathbb{L}}_{r+1}^0(\nabla G).$$

Theorem 17 establishes the following diagram of polytope inclusions and affine bijections:

$$
\begin{array}{ccccccc}
\text{For } M = M_0: & \mathbb{L}_{r+1}(G) & \subseteq & \text{Unnamed} & \subseteq & \mathbb{L}_r(G) & \\
& {\scriptstyle \textbf{Uproot}} \downarrow \uparrow {\scriptstyle \textbf{RootAt0}} & & {\scriptstyle \textbf{Uproot}} \downarrow \uparrow {\scriptstyle \textbf{RootAt0}} & & {\scriptstyle \textbf{Uproot}} \downarrow \uparrow {\scriptstyle \textbf{RootAt0}} & \quad (4) \\
\text{For } M^+: & \widetilde{\mathbb{L}}_{r+2}^0(\nabla G) & \subseteq & \widetilde{\mathbb{L}}_{r+1}(\nabla G) & \subseteq & \widetilde{\mathbb{L}}_{r+1}^0(\nabla G). &
\end{array}
$$

A question of theoretical interest and practical importance is which of the inclusions in (4) are strict. Our perspective here generalizes earlier work. Using different language, Deza and Laurent [2] identified $\mathbb{L}_2(G)$ with $\widetilde{\mathbb{L}}_3^0(\nabla G)$, which was termed RMET, the *rooted semimetric polytope*; and $\widetilde{\mathbb{L}}_3(\nabla G)$ with MET, the *semimetric polytope*. Building on this, Weller [19] considered $\mathbb{L}_3(G)$, the triplet-consistent polytope or TRI, though only in the context of pairwise potentials, and showed that $\mathbb{L}_3(G)$ has the remarkable property that if it is used to optimize an LP for a model $M$ on $G$, the exact same optimum is achieved for $\mathbb{L}_3(G_i)$ for any rerooting $M_i$. It was natural to conjecture that $\mathbb{L}_r(G)$ might have this same property for all $r > 3$, yet this was left as an open question.

## 6.2  $\mathbb{L}_3$ is unique in being universally rooted

We shall first strengthen [19] to show that $\mathbb{L}_3$ is *universally rooted* in the following stronger sense.

**Definition 18.** We say that the $r^{\text{th}}$-order Sherali-Adams relaxation is *universally rooted* (and write "$\mathbb{L}_r$ is universally rooted" for short) if for all admissible hypergraphs $G$, there is an affine score-preserving bijection between $\mathbb{L}_r(G)$ and $\mathbb{L}_r(G_i)$, for each rerooted hypergraph $(G_i)_{i \in V}$.

If $\mathbb{L}_r$ is universally rooted, this applies for potentials over up to $r$ variables (the maximum which makes sense in this context), and clearly it implies that optimizing score over any rerooting (as in MAP inference) will attain the same objective. The following result is proved in the Appendix §9.3.

**Lemma 19.** *If $\mathbb{L}_r$ is universally rooted for hypergraphs of maximum hyperedge degree $p < r$ with $p$ even, then $\mathbb{L}_r$ is also universally rooted for $r$-admissible hypergraphs with maximum degree $p + 1$.*

The proof relies on mapping to the symmetrized uprooted polytope $\widetilde{\mathbb{L}}_{r+1}^0(\nabla G)$. Then by considering marginals using a basis equivalent to that described in Proposition 11 for even $k$-potentials, we observe that the symmetry of the polytope enforces only one possible marginal for $(p + 1)$-clusters.

Combining Lemma 19 with arguments which extend those used by [19] demonstrates the following result, proved in the Appendix.

**Theorem 20.** $\mathbb{L}_3$ *is universally rooted.*

We next provide a striking and rather surprising result, see the Appendix for proof and details.

**Theorem 21.** $\mathbb{L}_3$ *is unique in being universally rooted. Specifically, for any integer $r > 1$ other than $r = 3$, we constructively demonstrate a hypergraph $G(V, E)$ with $|V| = r + 1$ variables for which $\widetilde{\mathbb{L}}_{r+1}^0(\nabla G) \neq \widetilde{\mathbb{L}}_{r+1}^i(\nabla G)$ for any $i \in V$.*

Theorem 21 examines $\widetilde{\mathbb{L}}_{r+1}^0(\nabla G)$ and $\widetilde{\mathbb{L}}_{r+1}^i(\nabla G)$, which by Theorem 17 are the uprooted equivalents of $\mathbb{L}_r(G)$ and $\mathbb{L}_r(G_i)$. It might appear more satisfying to try to demonstrate the result directly for the rooted polytopes, i.e. to show $\mathbb{L}_r(G) \neq \mathbb{L}_r(G_i)$. However, in general the rooted polytopes are not comparable: an $r$-potential in $M$ can map to an $(r + 1)$-potential in $M^+$ and then to an $(r + 1)$-potential in $M_i$ which cannot be evaluated for an $\mathbb{L}_r$ polytope.

Theorem 21 shows that we may hope for benefits from rerooting for any inference method based on a Sherali-Adams relaxed polytope $\mathbb{L}_r$, unless $r = 3$.

## 7  Experiments

Here we show empirically the benefits of uprooting and rerooting for approximate inference methods in models with higher-order potentials. We introduce an efficient heuristic which can be used in practice to select a variable for rerooting, and demonstrate its effectiveness.

We compared performance after different rerootings of marginal inference (to guarantee convergence we used the double loop method of Heskes et al. [4], which relates to generalized belief propagation, 24) and MAP inference (using loopy belief propagation, LBP [9]). For true values, we used the junction tree algorithm. All methods were implemented using libDAI [8]. We ran experiments on complete hypergraphs (with 8 variables) and toroidal grid models ($5 \times 5$ variables). Potentials up to order 4 were selected randomly, by drawing even $k$-potentials from $\mathrm{Unif}([-W_{\max}, W_{\max}])$ distributions for a variety of $W_{\max}$ parameters, as shown in Figure 2, which highlights results for estimating $\log Z$. For each regime of maximum potential values, we plot results averaged over 20 runs. For additional details and results, including marginals, other potential choices and larger models, see Appendix §10.

We display average error of the inference method applied to: the original model $M$; the uprooted model $M^+$; then rerootings at: the *worst* variable, the *best* variable, the *K heuristic* variable, and the *G heuristic* variable. *Best* and *worst* always refer to the variable at which rerooting gave with hindsight the best and worst error for the partition function (even in plots for other measures).

## 7.1 Heuristics to pick a good variable for rerooting

From our Definition 3, a rerooted model $M_i$ is obtained by clamping the uprooted model $M^+$ at variable $X_i$. Hence, selecting a good variable for rerooting is exactly the choice of a good variable to clamp in $M^+$. Considering pairwise models, Weller [19] refined the *maxW* method [20; 21] to introduce the *maxtW* heuristic, and showed that it was very effective empirically. *maxtW* selects the variable $X_i$ with $\max \sum_{j \in \mathcal{N}(i)} \tanh |\frac{W_{ij}}{4}|$, where $\mathcal{N}(i)$ is the set of neighbors of $i$ in the model graph, and $W_{ij}$ is the strength of the pairwise interaction.

The intuition for *maxtW* is as follows. Pairwise methods of approximate inference such as Bethe are exact for models with no cycles. If we could, we would like to 'break' tight cycles with strong edge weights, since these lead to error. When a variable is clamped, it is effectively removed from the model. Hence, we would like to reroot at a variable that sits on many cycles with strong edge weights. Identifying such cycles is NP-hard, but the *maxtW* heuristic attempts to do this by looking only locally around each variable. Further, the effect of a strong edge weight saturates [21]: a very strong edge weight $W_{ij}$ effectively 'locks' its end variables (either together or opposite depending on the sign of $W_{ij}$), and this effect cannot be significantly increased even by an extremely strong edge. Hence the $\tanh$ function was introduced to the earlier *maxW* method, leading to the *maxtW* heuristic.

As observed in §5, if we express our model potentials in terms of pure $k$-potentials, then the uprooted model will only have pure $k$-potentials for various values of $k$ which are even numbers. Intuitively, the higher the coefficients on these potentials, the more tightly connected is the model leading to more challenging inference. Hence, a natural way to generalize the *maxtW* approach to handle higher-order potentials is to pick a variable $X_i$ in $M^+$ which maximizes the following measure:

$$\text{clamp-heuristic-measure}(i) = \sum_{i \in \mathcal{E}: |\mathcal{E}|=2} c_2 \tanh |t_2 a_{\mathcal{E}}| + \sum_{i \in \mathcal{E}: |\mathcal{E}|=4} c_4 \tanh |t_4 a_{\mathcal{E}}|, \qquad (5)$$

where $a_{\mathcal{E}}$ is the coefficient (weight) of the relevant pure $k$-potential, see Definition 9, and the $\{c_2, t_2\}, \{c_4, t_4\}$ terms are constants for pure 2-potentials and for pure 4-potentials respectively. This approach extends to potentials of higher orders by adding similar further terms. Since our goal is to rank the measures for each $i \in V^+$, without loss of generality we take $c_2 = 1$. We fit the $t_2, c_4$ and $t_4$ constants to the data from our experimental runs, see the Appendix for details. Our *K heuristic* was fit only to runs for complete hypergraphs while the *G heuristic* was fit only to runs for models on grids.

## 7.2 Observations on results

Considering all results across models and approximate methods for estimating $\log Z$, marginals and MAP inference (see Figure 2 and Appendix §10.3), we make the following observations. Both K and G heuristics perform well (in and out of sample): they never hurt materially and often significantly improve accuracy, attaining results close to the best possible rerooting. Since our two heuristics achieve similar performance, sensitivity to the exact constants in (5) appears low. We verified this by comparing to maxtW for pairwise models as in [19]: both K and G heuristics performed just slightly better than maxtW. For all our runs, inference on rerooted models took similar time as on the original model (time required to reroot and later to map back inference results is negligible), see §10.3.1.

Observe that stronger 1-potentials tend to make inference *easier*, pulling each variable toward a specific setting, and reducing the benefits from rerooting (left column of Figure 2). Stronger pure $k$-potentials for $k > 1$ intertwine variables more tightly: this typically makes inference harder and increases the gains in accuracy from rerooting. The pure $k$-potential perspective facilitates this analysis.

When we examine larger models, or models with still higher order potentials, we observe qualitatively similar results, see Appendix §10.3.4 and 10.3.6.

## 8   Conclusion

We introduced methods which broaden the application of the uprooting and rerooting approach to binary models with higher-order potentials of any order. We demonstrated several important theoretical insights, including Theorems 20 and 21 which show that $\mathbb{L}_3$ is unique in being universally rooted. We developed the helpful tool of even $k$-potentials in §5, which may be of independent

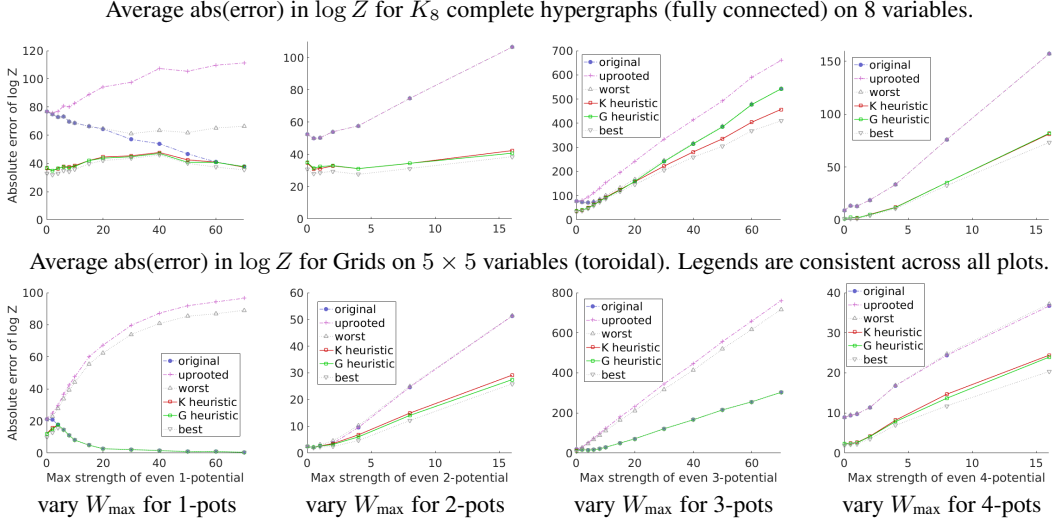

Average abs(error) in $\log Z$ for $K_8$ complete hypergraphs (fully connected) on 8 variables.

Average abs(error) in $\log Z$ for Grids on $5 \times 5$ variables (toroidal). Legends are consistent across all plots.

| vary $W_{\max}$ for 1-pots | vary $W_{\max}$ for 2-pots | vary $W_{\max}$ for 3-pots | vary $W_{\max}$ for 4-pots |

Figure 2: Error in estimating $\log Z$ for random models with various pure $k$-potentials over 20 runs. If not shown, $W_{\max}$ max coefficients for pure $k$-potentials are 0 for $k = 1$, 8 for $k = 2$, 0 for $k = 3$, 8 for $k = 4$. Where the red K heuristic curve is not visible, it coincides with the green G heuristic. Both K and G heuristics for selecting a rerooting work well: they never hurt and often yield large benefits. See §7 for details.

interest. We empirically demonstrated significant benefits for rerooting in higher-order models – particularly for the hard case of strong cluster potentials and weak 1-potentials – and provided an efficient heuristic to select a variable for rerooting. This heuristic is also useful to indicate when rerooting is unlikely to be helpful for a given model (if (5) is maximized by taking $i = 0$).

It is natural to compare the effect of rerooting $M$ to $M_i$, against simply clamping $X_i$ in the original model $M$. A key difference is that rerooting achieves the clamping at $X_i$ for negligible computational cost. In contrast, if $X_i$ is clamped in the original model then the inference method will have to be run twice: once clamping $X_i = 0$, and once clamping $X_i = 1$, then results must be combined. This is avoided with rerooting given the symmetry of $M^+$. Rerooting effectively replaces what may be a poor initial implicit choice of clamping at $X_0$ with a carefully selected choice of clamping variable almost for free. This is true even for large models where it may be advantageous to clamp a series of variables: by rerooting, one of the series is obtained for free, potentially gaining significant benefit with little work required. Note that each separate connected component may be handled independently, with its own added variable. This could be useful for (repeatedly) composing clamping and then rerooting each separated component to obtain an almost free clamping in each.

## Acknowledgements

We thank Aldo Pacchiano for helpful discussions, and the anonymous reviewers for helpful comments. MR acknowledges support by the UK Engineering and Physical Sciences Research Council (EPSRC) grant EP/L016516/1 for the University of Cambridge Centre for Doctoral Training, the Cambridge Centre for Analysis. AW acknowledges support by the Alan Turing Institute under the EPSRC grant EP/N510129/1, and by the Leverhulme Trust via the CFI.

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
