[Supplementary Material · genuproot-final-supp.pdf]

## APPENDIX: Uprooting and Rerooting Higher-Order Graphical Models

In this Appendix, we provide:

- In §9, proofs of results appearing in the main paper, split into:
  - §9.1 Proofs of results from §4: Recovery of inference tasks
  - §9.2 Proofs of results from §5: Even $k$-potentials
  - §9.3 Proofs of results from §6: Sherali-Adams relaxations.
- In §10, additional experimental details and results.

**Notation.** A model $M[G(V,E),(\theta_{\mathcal{E}})_{\mathcal{E}\in E}]$ uproots to $M^+[G^+(V^+,E^+,(\theta_{\mathcal{E}^+})_{\mathcal{E}^+\in E^+}]$, where $G^+ = \nabla G$. Given a model $M$ with hyperedges $\mathcal{E} \in E$ and potentials $(\theta_{\mathcal{E}})_{\mathcal{E}\in E}$, we adopt the convention that in the uprooted model $M^+$, each $\mathcal{E}^+ = \mathcal{E} \cup \{0\}$ and each $\theta_{\mathcal{E}^+}$ is the uprooted version of the respective $\theta_{\mathcal{E}}$, as given in Definition 2.

For a set $S$, we write $\#S = |S|$ for its cardinality. For example, $\#\{1,2,3\} = 3$.

## 9 Proofs of results from the main paper

### 9.1 Proofs of results from §4: Recovery of inference tasks

**Proposition 5 (Recovering the partition function)** Given a model $M[G(V,E),(\theta_{\mathcal{E}})_{\mathcal{E}\in E}]$ with partition function $Z$ as in (1), the partition function $Z^+$ of the uprooted model $M^+$ is twice $Z$, and the partition function of each rerooted model $M_i$ is exactly $Z$, for any $i \in V$.

*Proof.* Recall that for the model $M$, we have

$$Z = \sum_{x_V \in \{0,1\}^V} \exp\left(\sum_{\mathcal{E}\in E} \theta_{\mathcal{E}}(x_{\mathcal{E}})\right).$$

Writing $Z^+$ for the partition function of $M^+$, by definition we have

$$
\begin{aligned}
Z^+ &= \sum_{x_{V\cup\{0\}} \in \{0,1\}^{V\cup\{0\}}} \exp\left(\sum_{\mathcal{E}^+\in E^+} \theta_{\mathcal{E}^+}(x_{\mathcal{E}\cup\{0\}})\right) \\
&= \sum_{x_V \in \{0,1\}^V} \exp\left(\sum_{\mathcal{E}^+\in E^+} \theta_{\mathcal{E}^+}(x_0=0,x_{\mathcal{E}})\right) + \sum_{x_V \in \{0,1\}^V} \exp\left(\sum_{\mathcal{E}^+\in E^+} \theta_{\mathcal{E}^+}(x_0=1,x_{\mathcal{E}})\right) \\
&= \sum_{x_V \in \{0,1\}^V} \exp\left(\sum_{\mathcal{E}^+\in E^+} \theta_{\mathcal{E}^+}(x_0=0,x_{\mathcal{E}})\right) + \sum_{x_V \in \{0,1\}^V} \exp\left(\sum_{\mathcal{E}^+\in E^+} \theta_{\mathcal{E}^+}(x_0=1,x_{\mathcal{E}})\right) \\
&= \sum_{x_V \in \{0,1\}^V} \exp\left(\sum_{\mathcal{E}\in E} \theta_{\mathcal{E}}(x_{\mathcal{E}})\right) + \sum_{x_V \in \{0,1\}^V} \exp\left(\sum_{\mathcal{E}\in E} \theta_{\mathcal{E}}(\overline{x}_{\mathcal{E}})\right) \\
&= 2Z,
\end{aligned}
$$

as required. Now, given $i \in V$, and noting that $M^+$ is also the uprooting of the model $M_i$, it immediately follows from the above that the partition function associated with $M_i$ is $Z$, as required. $\square$

**Proposition 6 (Recovering MAP configurations)** From $M^+$: $x_V$ is an arg max for $p$ iff $(x_0 = 0, x_V)$ is an arg max for $p^+$ iff $(x_0 = 1, \overline{x}_V)$ is an arg max for $p^+$. From a rerooted model $M_i$: $(x_{V\setminus\{i\}}, x_i = 0)$ is an arg max for $p$ iff $(x_0 = 0, x_{V\setminus\{i\}})$ is an arg max for $p_i$; $(x_{V\setminus\{i\}}, x_i = 1)$ is an arg max for $p$ iff $(x_0 = 1, \overline{x}_{V\setminus\{i\}})$ is an arg max for $p_i$.

*Proof.* From $M^+$: we simply note that by construction of the uprooted potentials, for any $x_V \in \{0,1\}^V$ we have

$$\sum_{E\in E} \theta_E(x_E) = \sum_{E\in E} \theta_{E^+}^+(x_E, x_0=0) = \sum_{E\in E} \theta_{E^+}^+(\overline{x}_E, x_0=1),$$

from which the claim immediately follows.

From $M_i$: we have

$$p_i(x_{V\setminus\{i\}}, x_0) \propto p^+(x_{V\setminus\{i\}}, x_0, x_i = 0),$$

which implies that

$$
\begin{aligned}
(x_{V\cup\{0\}\setminus\{i\}}) \in \arg\max p_i \iff & \ (x_{V\cup\{0\}\setminus\{i\}}, x_i = 0) \in \arg\max p^+ \\
\iff & \ (\overline{x}_{V\cup\{0\}\setminus\{i\}}, x_i = 1) \in \arg\max p^+ \\
\iff & \ \begin{cases} (x_{V\setminus\{i\}}, x_i = 0) \in \arg\max p & \text{if } x_0 = 0 \\ (\overline{x}_{V\setminus\{i\}}, x_i = 1) \in \arg\max p & \text{if } x_0 = 1. \end{cases}
\end{aligned}
$$

$\square$

**Proposition 7 (Recovering marginals)**  For a subset $\emptyset \neq U \subseteq V$, we can recover from $M^+$:

$$p(x_U) = p^+(x_0 = 0, x_U) + p^+(x_0 = 1, \overline{x}_U) \qquad = 2p^+(x_0 = 0, x_U) = 2p^+(x_0 = 1, \overline{x}_U).$$

To recover from a rerooted $M_i$: (i) For any $i \in V \setminus U$, $\ p(x_U) = p_i(x_0 = 0, x_U) + p_i(x_0 = 1, \overline{x}_U)$.
(ii) For any $i \in U$, $p(x_U) = \begin{cases} p_i(x_0 = 0, x_{U\setminus\{i\}}) & x_i = 0 \\ p_i(x_0 = 1, \overline{x}_{U\setminus\{i\}}) & x_i = 1. \end{cases}$

*Proof.* Let $x_U \in \{0,1\}^U$. Observe that

$$
\begin{aligned}
p(x_U) &= \frac{1}{Z} \sum_{x_{V\setminus U}} \exp\left(\sum_{\mathcal{E} \in E} \theta_{\mathcal{E}}(x_{\mathcal{E}})\right) \\
&= \frac{1}{Z} \sum_{x_{V\setminus U}} \exp\left(\sum_{\mathcal{E}^+ \in E^+} \theta_{\mathcal{E}^+}(x_0 = 0, x_{\mathcal{E}})\right) \\
&= \frac{1}{2Z}\left(\sum_{x_{V\setminus U}} \exp\left(\sum_{\mathcal{E}^+ \in E^+} \theta_{\mathcal{E}^+}(x_0 = 0, x_{\mathcal{E}})\right) + \sum_{x_{V\setminus U}} \exp\left(\sum_{\mathcal{E}^+ \in E^+} \theta_{\mathcal{E}^+}(x_0 = 1, \overline{x}_{\mathcal{E}})\right)\right) \\
&= p^+(x_0 = 0, x_U) + p^+(x_0 = 1, \overline{x}_U) = 2p^+(x_0 = 0, x_U) = 2p^+(x_0 = 1, \overline{x}_U).
\end{aligned}
$$

We next demonstrate recovery from a rerooted model $M_i$. Let $V_i = V \cup \{0\} \setminus \{i\}$. By the definition of rerooting and symmetry of $M^+$, $\ p_i(x_{V_i}) = p^+(x_{V_i}|x_i = 0) = p^+(\overline{x}_{V_i}|x_i = 1)$. Further, $p^+(x_i = 0) = p^+(x_i = 1) = \frac{1}{2}$ for any $i = 0, 1, \ldots, n$.

Case (i) $i \in V \setminus U$. Following the argument above, we obtain

$$
\begin{aligned}
p(x_U) &= p^+(x_0 = 0, x_U) + p^+(x_0 = 1, \overline{x_U}) \\
&= p^+(x_0 = 0, x_U, x_i = 0) + p^+(x_0 = 0, x_U, x_i = 1) \\
&\quad + p^+(x_0 = 1, \overline{x_U}, x_i = 0) + p^+(x_0 = 1, \overline{x_U}, x_i = 1) \\
&= \frac{1}{2}\left[p^+(x_0 = 0, x_U|x_i = 0) + p^+(x_0 = 0, x_U|x_i = 1)\right] \\
&\quad + \frac{1}{2}\left[p^+(x_0 = 1, \overline{x}_U|x_i = 0) + p^+(x_0 = 1, \overline{x}_U|x_i = 1)\right] \\
&= \frac{1}{2}\left[p^+(x_0 = 0, x_U|x_i = 0) + p^+(x_0 = 1, \overline{x}_U|x_i = 1)\right] \\
&\quad + \frac{1}{2}\left[p^+(x_0 = 0, x_U|x_i = 1) + p^+(x_0 = 1, \overline{x}_U|x_i = 0)\right] \\
&= p_i(x_0 = 0, x_U) + p_i(x_0 = 1, \overline{x}_U).
\end{aligned}
$$

Case (ii) $i \in U$. Now we have

$$
\begin{aligned}
p(x_U) &= p^+(x_0 = 0, x_U) + p^+(x_0 = 1, \overline{x}_U) \\
&= \frac{1}{2}\left[p^+(x_0 = 0, x_U|x_i = 0) + p^+(x_0 = 0, x_U|x_i = 1)\right]
\end{aligned}
$$

$$+ \frac{1}{2} \left[ p^+(x_0 = 1, \overline{x}_U | x_i = 0) + p^+(x_0 = 1, \overline{x}_U | x_i = 1) \right]$$

$$= \begin{cases} p_i(x_0 = 0, x_{U \setminus \{i\}}) & \text{if } x_i = 0 \\ p_i(x_0 = 1, \overline{x}_{U \setminus \{i\}}) & \text{if } x_i = 1. \end{cases} \qquad \square$$

## 9.2 Proofs of results from §5: Even $k$-potentials

**Proposition 10 (All pure potentials are essentially even potentials)** Let $k \geq 2$, and $|U| = k$. If $\theta_U : \{0,1\}^U \to \mathbb{R}$ is a pure $k$-potential then $\theta_U$ must be an affine function of the even $k$-potential, i.e. $\exists\, a, b \in \mathbb{R}$ s.t. $\theta_U(x_U) = a \mathbb{1}[\, |\{i \in U | x_i = 1\}| \text{ is even}] + b$.

*Proof.* It is sufficient to demonstrate that if, for two configurations $x_U, y_U \in \{0,1\}^U$, we have $\sum_{i \in U} x_i = \sum_{i \in U} y_i \bmod 2$, then $\theta_U(x_U) = \theta_U(y_U)$, since this demonstrates that $\theta_U$ depends on its input argument only through the quantity $\mathbb{1}_{\#\{i \in U | x_i = 1\} \text{ is even}}$, and since this only takes on two possible values, $\theta_U$ may be expressed as an affine function of this indicator.

To demonstrate the claim above, it is sufficient to show that if $x_U \in \{0,1\}^U$, and $i, j \in V$ are two distinct indices, and $F_{ij}(x_U) \in \{0,1\}^U$ denotes the configuration obtained from $x_U$ by flipping coordinates $i$ and $j$, then $\theta_U(x_U) = \theta_U(F_{ij}(x_U))$. This is sufficient since given $x_U, y_U \in \{0,1\}^U$ with $\sum_{i \in U} x_i = \sum_{i \in U} y_i \bmod 2$, it is possible to obtain $y_U$ from $x_U$ by iteratively flipping pairs of distinct variables.

Let $F_i(x_U)$ denote the configuration obtained from $x_U$ by flipping $x_i$. By the uniform marginalization property, we have

$$p(x_U) + p(F_i(x_U)) = p(F_j(x_U)) + p(F_{ij}(x_U))$$

and

$$p(F_i(x_U)) + p(F_{ij}(x_U)) = p(x_U) + p(F_j(x_U)) \,.$$

Subtracting these equations from one another yields

$$p(x_U) = p(F_{ij}(x_U)) \,.$$

Taking logarithms of this equations yields $\theta_U(x_U) = \theta_U(F_{ij}(x_U))$, as required. $\qquad \square$

**Proposition 11 (Even $k$-potentials form a basis)** For a finite set $U$, the set of even $k$-potentials $\left( \mathbb{1}[\, |\{i \in W | X_i = 1\}| \text{ is even}] \right)_{W \subseteq U}$, indexed by subsets $W \subseteq U$, forms a basis for the vector space of all potential functions $\theta : \{0,1\}^U \to \mathbb{R}$.

*Proof.* We show that the indicators $(\mathbb{1}[\#\{i \in W | x_i = 1\} \text{ is even}])_{W \subseteq U}$ form a basis for the vector space of functions $\mathbb{R}^{\{0,1\}^U}$; we interpret the indicator corresponding to the empty set as being the constant function equal to 1. Given this, we then note that $\mathscr{P}(\{0,1\}^U)$ is a convex subset of an affine subspace of $\mathbb{R}^{\{0,1\}^U}$ of co-dimension 1, and that the indicator corresponding to the empty set is orthogonal to this affine subspace. This is then sufficient to show that for any probability distribution $\mu \in \mathscr{P}(\{0,1\}^U)$, there is a unique set of parameters $(\eta_W)_{\emptyset \neq W \subseteq U}$ such that

$$\mu(x_U) = \sum_{\emptyset \neq W \subseteq U} \eta_W \, \mathbb{1}[\#\{i \in W | x_i = 1\} \text{ is even}] \,,$$

as required.

To demonstrate that $(\mathbb{1}[\#\{i \in W | x_i = 1\} \text{ is even}])_{W \subseteq U}$ form a basis for the vector space of functions $\mathbb{R}^{\{0,1\}^U}$, we first note that it has the correct number of elements to form a basis, and it is therefore sufficient to either demonstrate that it is a spanning set, or that it is a linearly independent set; we take the latter approach.

Suppose we have a collection of coefficients $(\alpha_W)_{W \subseteq U}$ such that

$$\sum_{W \subseteq U} \alpha_W \, \mathbb{1}[\#\{i \in W | x_i = 1\} \text{ is even}] = 0 \,.$$

Given a subset $X \subseteq U$, note that we have

$$\Big(\mathbb{1}[\#\{i \in X | x_i = 1\} \text{ is even}] - \mathbb{1}[\#\{i \in X | x_i = 1\} \text{ is odd}]\Big) \cdot$$

$$\Big(\sum_{W \subseteq U} \alpha_W \mathbb{1}[\#\{i \in W | x_i = 1\} \text{ is even}]\Big) = 0$$

$$\implies \sum_{W \subseteq U} \alpha_W \sum_{x \in \{0,1\}^U} \Big(\mathbb{1}[\#\{i \in W | x_i = 1\} \text{ is even}, \#\{i \in X | x_i = 1\} \text{ is even}]$$

$$- \mathbb{1}[\#\{i \in W | x_i = 1\} \text{ is even}, \#\{i \in X | x_i = 1\} \text{ is odd}]\Big) = 0. \tag{6}$$

Considering the summand above for a fixed subset $W \subseteq U$, note that if $W = X$, then the result of summing over all configurations $x_U \in \{0,1\}^U$ is $2^{|U|-1}$. However, if $W \neq X$, the result of the sum is 0. From this it immediately follows that $\alpha_X = 0$, and the proof of linear independence is complete. An elegant perspective which demonstrates that the sum concerned above evaluates to 0 is to view $\{0,1\}^U$ as a vector space over the finite field with 2 elements $\mathbb{F}_2$, with addition defined componentwise. In this case, the set $\{x \in \{0,1\}^U | \#\{i \in W | x_i = 1\}\}$ is exactly the kernel of the linear form $\{0,1\}^U \ni x \mapsto \sum_{i \in W} x_i \in \mathbb{F}_2$ (where the addition is to be interpreted modulo 2). Considering the linear form $\{x \in \{0,1\}^U | \#\{i \in W | x_i = 1\}\} \ni x \mapsto \sum_{i \in X} x_i \in \mathbb{F}_2$, we observe that the two sets

$$\{x \in \{0,1\}^U | \#\{i \in W | x_i = 1\} \text{ is even}, \#\{i \in X | x_i = 1\} \text{ is even}\} \text{ and}$$
$$\{x \in \{0,1\}^U | \#\{i \in W | x_i = 1\} \text{ is even}, \#\{i \in X | x_i = 1\} \text{ is odd}\},$$

are the preimage of $0 \in \mathbb{F}_2$ and $1 \in \mathbb{F}_2$ under this linear form, respectively. Therefore, if the linear form is surjective, the two sets have the same size, and since they are clearly disjoint, the relevant term of (6) evaluates to 0. To see that the form is surjective, recall that by assumption $X \neq W$. If $X \setminus W$ is non-empty, then surjectivity is demonstrated by changing a single coordinate corresponding to an index in $X \setminus W$. If $X \setminus W$ is empty, then $W \setminus X$ is non-empty, and by simultaneously chaning a coordinate in $W \setminus X$ and $X$, surjectivity is demonstrated. □

### 9.3 Proofs of results from §6: Sherali-Adams relaxations

**Theorem 17.** For a hypergraph $G = (V, E)$, and integer $r$ such that $\max_{\mathcal{E} \in E} |\mathcal{E}| \leq r \leq |V|$, there is an affine score-preserving bijection

$$\mathbb{L}_r(G) \overset{\textbf{Uproot}}{\underset{\textbf{RootAt0}}{\rightleftarrows}} \widetilde{\mathbb{L}}^0_{r+1}(\nabla G) \,.$$

*Proof.* The structure of the proof is as follows. We first construct the uprooting map **Uproot**, which we will denote by $\Psi : \mathbb{L}_k(G) \to \widetilde{\mathbb{L}}^0_{k+1}(\nabla G)$ for notational convenience, and show that it is bijective by exhibiting its double-sided inverse, **RootAt**0, which we will denote by $\Phi : \widetilde{\mathbb{L}}^0_{k+1}(\nabla G) \to \mathbb{L}_k(G)$. We then directly show that this bijection is affine and score-preserving.

To construct $\Psi$, let $\boldsymbol{\mu} \in \mathbb{L}_k(G)$, and define

$$\Psi(\boldsymbol{\mu}) = \boldsymbol{\mu}^+ = (\mu_U^+)_{\substack{U \subseteq V \\ |U \setminus \{0\}| \leq k}} \in \widetilde{\mathbb{L}}^0_{k+1}(\nabla G)$$

as follows. We begin defining the measures $\mu_U^+$ for subsets $U$ not including the additional element $0 \in V^+$ in the suspension graph; let $U \subseteq V$ with $|U| \leq k$. We define the 'symmetrized' measures

$$\mu_U^+(x_U) = \frac{1}{2}\big[\mu_U(x_U) + \mu_U(\overline{x}_U)\big] \qquad \forall x_U \in \{0,1\}^U. \tag{7}$$

Now turning our attention to subsets that *do* contain the new element $0 \in V^+$, we write $U^+ = U \cup \{0\}$, and define:

$$\mu_{U^+}^+(x_{U^+}) = \begin{cases} \frac{1}{2}\mu_U(x_U) & \text{if } x_0 = 0 \\ \frac{1}{2}\mu_U(\overline{x}_U) & \text{if } x_0 = 1 \end{cases} \qquad \forall x_{U^+} \in \{0,1\}^{U^+}. \tag{8}$$

We define $\mu_{\{0\}}(x_0)$ to take value $1/2$ for $x_0 = 0$ and $x_0 = 1$. We have now defined the entire collection of probability measures $\boldsymbol{\mu}^+$. Note that by construction, each individual measure in the collection is flipping-invariant, and by observing the form of Equations (7) and (8), we observe that the map is affine. We now demonstrate consistency of these measures. Let $W \subset U \subseteq V \cup \{0\}$. We aim to demonstrate

$$\mu_W^+(x_W) = \sum_{\substack{y_U \in \{0,1\}^U \\ y_W = x_W}} \mu_U^+(y_U) \tag{9}$$

There are three cases to consider: (i) $W \not\subseteq V$ (i.e. both subsets contain 0), (ii) $U \subseteq V$ (i.e. neither subset contains 0), (iii) $0 \in U, 0 \notin W$. In the first two cases, the marginalization consistency condition of Equation (9) follows immediately from the definitions in Equations (7) and (8), and recalling the consistency of the collection of measures $\boldsymbol{\mu}$. For case (iii), we write $U = A \cup \{0\}$ for $A \subset V$ and directly calculate:

$$\sum_{\substack{y_U \in \{0,1\}^U \\ y_W = x_W}} \mu_U^+(y_U) = \sum_{\substack{y_U \in \{0,1\}^U \\ y_W = x_W \\ y_0 = 0}} \frac{1}{2} \mu_A(y_A) + \sum_{\substack{y_U \in \{0,1\}^U \\ y_W = x_W \\ y_0 = 1}} \frac{1}{2} \mu_A(F_A(y_A))$$

$$= \sum_{\substack{y_A \in \{0,1\}^A \\ y_W = x_W}} \frac{1}{2} \mu_A(y_A) + \sum_{\substack{y_A \in \{0,1\}^A \\ y_W = x_W}} \frac{1}{2} \mu_A(F_A(y_A))$$

$$= \sum_{\substack{y_A \in \{0,1\}^A \\ y_W = x_W}} \mu_A^+(y_A),$$

so $\mu_U^+$ and $\mu_A^+$ are consistent. The consistency of $\mu_U^+$ and $\mu_W^+$ then follows from case (ii). Having checked consistency, we have verified that the map $\Psi : \mathbb{L}_k(G) \to \widetilde{\mathbb{L}}_{k+1}^0(\nabla G)$ is well-defined. We now exhibit its inverse. Given $\boldsymbol{\eta} \in \widetilde{\mathbb{L}}_{k+1}^0(\nabla G)$, we define $\Phi(\boldsymbol{\eta}) = \boldsymbol{\mu} = (\mu_U)_{|U| \leq k} \in \mathbb{L}_k(G)$ as follows. Given $U \subseteq V, |U| \leq k$, write $U^+ = U \cup \{0\}$, and define

$$\mu_U(x_U) = \eta_{U^+}(x_0 = 0, x_U) + \eta_{U^+}(x_0 = 1, \overline{x}_U)$$

We now directly show that for $\boldsymbol{\mu} \in \mathbb{L}_k(G)$, we have $\Phi(\Psi(\boldsymbol{\mu})) = \boldsymbol{\mu}$. We take $|U| \leq k$, and note that from our definitions of $\Psi$ and $\Phi$, we have for all $x_U \in \{0,1\}^U$ that

$$\Phi(\Psi(\boldsymbol{\mu}))_U(x_U) = \mu_U^+(x_U, x_0 = 0) + \mu_U^+(\overline{x}_U, x_0 = 1) = \frac{1}{2}\mu_U(x_U) + \frac{1}{2}\mu_U(\overline{x}_U) = \mu_U(x_U).$$

Now let $\boldsymbol{\mu} \in \widetilde{\mathbb{L}}_{k+1}^0(G)$. We demonstrate that $\boldsymbol{\mu}'' = \Psi(\Phi(\boldsymbol{\mu})) = \boldsymbol{\mu}$. First, for $U \subseteq V, |U| \leq k$, we have

$$\Psi(\Phi(\boldsymbol{\mu}))_U(x_U) = \frac{1}{2}\left(\mu_U^+(x_U) + \mu_U^+(\overline{x}_U)\right)$$

$$= \frac{1}{2}\Big(\mu_{U\cup\{0\}}(x_U, x_0 = 0) + \mu_{U\cup\{0\}}(\overline{x}_U, x_0 = 1)$$

$$\qquad + \mu_{U\cup\{0\}}(\overline{x}_U, x_0 = 0) + \mu_{U\cup\{0\}}(x_U, x_0 = 1)\Big)$$

$$= \frac{1}{2}\left(\mu_U(x_U) + \mu_U(\overline{x}_U)\right)$$

$$= \mu_U(x_U),$$

where in the final equality we have used the flipping-invariance of $\mu_U$. Secondly, for $U \subseteq V$, write $U^+ = U \cup \{0\}$, and note

$$\Psi(\Phi(\boldsymbol{\mu}))_{U^+}(x_{U^+}) = \frac{1}{2}\mu_U^+(x_U)\mathbb{1}[x_0 = 0] + \frac{1}{2}\mu_U^+(\overline{x}_U)\mathbb{1}[x_0 = 1]$$

$$= \frac{1}{2}\left(\mu_{U^+}(x_U, x_0 = 0) + \mu_{U^+}(\overline{x}_U, x_0 = 1)\right)\mathbb{1}[x_0 = 0]$$

$$\qquad + \frac{1}{2}\left(\mu_{U^+}(\overline{x}_U, x_0 = 0) + \mu_{U^+}(\overline{\overline{x}}_U), x_0 = 1)\right)\mathbb{1}[x_0 = 1]$$

$$= \frac{1}{2}\left(\mu_{U^+}(x_U^+) + \mu_{U^+}(\overline{x}_{U^+})\right)\mathbb{1}[x_0 = 0]$$

$$+ \frac{1}{2}\left(\mu_{U^+}(\overline{x}_{U^+}) + \mu_{U^+}(x_{U^+}))\right)\mathbb{1}[x_0 = 1]$$

$$= \mu_{U^+}(x_{U^+}),$$

where again in the final equality we have used the flipping-invariance of $\mu_{U^+}$.

Finally, to see that the map is score-preserving, let $(\theta_{\mathcal{E}})_{\mathcal{E} \in E}$ be a collection of potentials defining a model on $G = (V, E)$. Then for any $\boldsymbol{\mu}^+ \in \mathbb{L}_{k+1}^0(G)$, note that we have

$$\sum_{\mathcal{E} \in E} \mathbb{E}_{X_{\mathcal{E} \cup \{0\}} \sim \mu_{\mathcal{E} \cup \{0\}}^+}\left[\theta_{\mathcal{E} \cup \{0\}}(X_{\mathcal{E} \cup \{0\}})\right]$$

$$= \sum_{\mathcal{E} \in E} \sum_{x_{\mathcal{E}^+} \in \{0,1\}^{\mathcal{E}^+}} \theta_{C^+}(x_{C^+})\mu_{\mathcal{E}^+}^+(x_{\mathcal{E}^+})$$

$$= \sum_{\mathcal{E} \in E}\left[\sum_{\substack{x_{\mathcal{E}^+} \in \{0,1\}^{\mathcal{E}^+} \\ x_0 = 0}} \theta_{\mathcal{E}^+}(x_{\mathcal{E}^+})\mu_{\mathcal{E}^+}^+(x_{\mathcal{E}^+}) + \sum_{\substack{x_{\mathcal{E}^+} \in \{0,1\}^{\mathcal{E}^+} \\ x_0 = 1}} \theta_{\mathcal{E}^+}(x_{\mathcal{E}^+})\mu_{\mathcal{E}^+}^+(x_{\mathcal{E}^+})\right]$$

$$= \sum_{\mathcal{E} \in E}\left[\sum_{\substack{x_{\mathcal{E}^+} \in \{0,1\}^{\mathcal{E}^+} \\ x_0 = 0}} \theta_{\mathcal{E}}(x_{\mathcal{E}})\frac{1}{2}\mu_{\mathcal{E}}(x_{\mathcal{E}}) + \sum_{\substack{x_{\mathcal{E}^+} \in \{0,1\}^{\mathcal{E}^+} \\ x_0 = 1}} \theta_{\mathcal{E}}(\overline{x}_{\mathcal{E}})\frac{1}{2}\mu_{\mathcal{E}}(\overline{x}_{\mathcal{E}})\right]$$

$$= \sum_{\mathcal{E} \in E} \sum_{x_C \in \{0,1\}^C} \theta_{\mathcal{E}}(x_{\mathcal{E}})\mu_{\mathcal{E}}(x_{\mathcal{E}})$$

$$= \sum_{\mathcal{E} \in E} \mathbb{E}_{X_{\mathcal{E}} \sim \mu_{\mathcal{E}}}\left[\theta_{\mathcal{E}}(X_{\mathcal{E}})\right],$$

as required.

$\square$

**Lemma 19.** If $\mathbb{L}_r$ is universally rooted for hypergraphs of maximum hyperedge degree $p < r$ with $p$ even, then $\mathbb{L}_r$ is also universally rooted for $r$-admissible hypergraphs with maximum degree $p + 1$.

*Proof.* The key observation is that given some set of variables $x_U$ of size $p + 1$, if we have a set of flipping-invariant probability measures on $\{0, 1\}^W$ for each subset $W \subseteq U$ of size $p$ which are consistent, then by Proposition 11, then a flipping-invariant probability measure over $\{0, 1\}^U$ is specified by one additional parameter. The parameter corresponds to the even potential $U$, and is given by

$$\mathbb{P}(|\{i \in U | x_i = 1\}| \text{ is even})$$

But since $p + 1$ is odd, and we require the measure to be flipping-invariant, the only possible value for this parameter must be $1/2$. Moreover, taking the parameter to be $1/2$ must yield a valid distribution over $\{0, 1\}^U$, as we assumed that the measures on each of $\{0, 1\}^W$ ($W \subseteq U$, $|W| = p$) were consistent.

This demonstrates that given a hypergraph $G$ with maximum hyperedge degree $p + 1$, we can construct a new hypergraph $G' = (V, E')$, with the same vertex set as $G$, and hyperedge set defined by

$$E' = \{\mathcal{E} \in E | |\mathcal{E}| \le p\} \cup \{U \subset V | U \subseteq \mathcal{E} \in E, |E| = p + 1, |U| = p\}$$

From our argument above, we have $\mathbb{L}_r(G)$ is in affine bijection with $\mathbb{L}_r(G')$, and since $G'$ has maximum hyperedge degree $p$, the statement of the lemma follows. $\square$

**Theorem 20.** $\mathbb{L}_3$ is universally rooted.

*Proof.* We observe that it is straightforward to extend the analysis in the Appendix of [19] to demonstrate that for any hypergraph of maximum hyperedge degree 2, there exists a score-preserving affine bijection between $\mathbb{L}_3(G)$ and each of its rerootings. We now combine this with the observation of Lemma 19, taking $p = 2$, from which the statement of the theorem immediately follows. $\qquad\square$

**Theorem 21.** $\mathbb{L}_3$ is unique in being universally rooted. Specifically, for any integer $r > 1$ other than $r = 3$, we constructively demonstrate a hypergraph $G = (V, E)$ with $|V| = r + 1$ variables for which $\widetilde{\mathbb{L}}^0_{r+1}(\nabla G) \neq \widetilde{\mathbb{L}}^i_{r+1}(\nabla G)$ for any $i \in V$.

*Proof.* For each $k \neq 3$, we shall constructively demonstrate a model $M$ on hypergraph $G$ as stated such that the LP relaxation over $\mathbb{L}_k(G)$ is not tight for $M$ but the LP relaxation over $L_k(\nabla G \setminus \{i\})$ is tight for every rerooted model $M_i, i \in V$.

**Case 1: $k$ is even.** Let $G = (V, E)$, with $V = \{1, \ldots, k + 1\}$, and $E$ the set of all subsets of $V$ of size $k$. Consider a model with the following set of potentials on this hypergraph:

$$\theta_{\mathcal{E}}(x_{\mathcal{E}}) = -\mathbb{1}[\#\{i \in \mathcal{E} | x_i = 1\} \text{ is even}] \qquad \forall \mathcal{E} \in E. \tag{10}$$

The optimum score for a configuration $x_V \in \{0, 1\}^V$ is $-1$. We show this by demonstrating (i) that the optimum is at most -1 (which is all we need here), then (ii) that the optimum is at least -1. For (i): Toward contradiction, assume that there exists a configuration that does not activate any of the $\theta_{\mathcal{E}}$ potentials, i.e. all $k$-clusters have an odd number of 1s. Pick one of the $k$-clusters and call it $S$. Since $k \geq 2$ is even, $S$ contains at least one variable set to 0, call it $x$, and at least one set to 1, call it $y$. Now $V$ has $k + 1$ variables consisting of $S$ together with one more variable $z$. If $z = 0$ then consider the $k$-cluster $T = S \setminus \{y\} \cup \{z\}$. If $z = 1$ then let $T = S \setminus \{x\} \cup \{z\}$. In either case, $T$ has an even number of 1s, contradiction. For (ii): Consider the setting $x_1 = 1$ with all other variables set to 0. All $k$-clusters including $x_1$ are inactive. There is just one $k$-cluster not including $x_1$, and this $k$-cluster has no 1s thus its potential is active. Hence, this configuration achieves a score of $-1$.

However, the set of pseudomarginal distributions in $\mathbb{L}_k(G)$ below attains a score of 0:

$$\mu_{\mathcal{E}}(x_{\mathcal{E}}) = \frac{1}{k} \sum_{i \in \mathcal{E}} \delta_{x_i = 1, x_{\mathcal{E} \setminus \{i\}} = 0} \qquad \forall \mathcal{E} \in E.$$

Now observe that when this model is uprooted, we have the hypergraph $\nabla G = (V^+, E)$, where $V^+ = \{0\} \cup V$, and the hyperedge set $E^+ = E$ as before with the same set of potentials as in (10), by Lemma 12. Therefore, rerooting at a variable $i \in \{1, \ldots, k + 1\}$ will result in a graphical model on the graph $\nabla G \setminus \{i\}$ with vertices $\{0, 1, \ldots, k + 1\} \setminus \{i\}$, and hyperedges given by one hyperedge of size $k$ (the original hyperedge which did not include $i$), which is $\{1, \ldots, k + 1\} \setminus \{i\}$, along with all subsets of $\{1, \ldots, k + 1\} \setminus \{i\}$ of size $k - 1$. In particular, the model consists of potentials over the set of $k$ variables $\{1, \ldots, k + 1\} \setminus \{i\}$, and the variable $X_0$ is independent from the rest of the variables, with symmetric distribution on its state space $\{0, 1\}$. Therefore, the polytope $\mathbb{L}_k(\nabla G \setminus \{i\})$ is tight for this potential since it is effectively a model over $k$ variables, proving the claim.

**Case 2: $k \geq 5$ is odd.** Let $k \geq 5$ be odd, and again let $G = (V, E)$, with $V = \{1, \ldots, k + 1\}$, this time letting $E$ be the set of all subsets of $V$ of size $k - 1$ (an even number). Consider the following set of potentials on this hypergraph

$$\theta_{\mathcal{E}}(x_{\mathcal{E}}) = -\mathbb{1}[\#\{i \in \mathcal{E} | x_i = 1\} \text{ is even}] \qquad \forall \mathcal{E} \in E.$$

We note that the polytope $\mathbb{L}_k(G)$ is not tight for this polytope, by considering the following set of pseudomarginals over hyperedges of $G$:

$$\mu_{\mathcal{E}}(x_{\mathcal{E}}) = \frac{1}{k} \delta_{x_i = 0 \forall i \in \mathcal{E}} + \frac{1}{k} \sum_{i \in \mathcal{E}} \delta_{x_i = 1, x_{\mathcal{E} \setminus \{i\}} = 0} \qquad \forall \mathcal{E} \in E.$$

These are valid pseudomarginals in $\mathbb{L}_k(G)$, as the following distributions over $k$-clusters are consistent and marginalize down to the distributions over hyperedges:

$$\mu_U(x_U) = \frac{1}{k} \sum_{i \in U} \delta_{x_i = 1, x_{U \setminus \{i\}} = 0} \qquad \forall U \subseteq V, |U| = k.$$

Note that the score of this set of pseudomarginals is

$$\sum_{\mathcal{E} \in E} -\mu_{\mathcal{E}}(\#\{i \in \mathcal{E} | x_i = 1\} \text{ is even}) = -\binom{k+1}{k-1}\frac{1}{k} = -\frac{k+1}{2}$$

We now argue that this exceeds the maximum score obtainable by a configuration $x_V \in \{0,1\}^V$, demonstrating non-tightness of $\mathbb{L}_k(G)$ for this model. To see this, let $\ell \in \{0, \ldots, k+1\}$ be the number of non-zero coordinates of $x_V$. We count the number of subsets $U$ of $\{1, \ldots, k+1\}$ of size $k-1$ for which $x_U$ has an even number of non-zero coordinates, and show that this is greater than $(k+1)/2$, leading to a score less than $-(k+1)/2$. The number of such subsets is given by:

$$\sum_{p=0}^{\lfloor \ell/2 \rfloor} \binom{\ell}{2p}\binom{k+1-l}{k-1-2p} = \begin{cases} (k+1)(k-1)/2 & \ell = 0 \\ \binom{\ell}{\ell-2}\binom{k+1-l}{k+1-l} + \binom{\ell}{\ell}\binom{k+1-l}{k-1-\ell} = \frac{\ell(\ell-1)}{2} + \frac{(k+1-\ell)(k-\ell)}{2} & \ell \neq 0 \text{ even} \\ \binom{\ell}{\ell-1} + \binom{k+1-\ell}{k-\ell} = k+1 & \ell \text{ odd.} \end{cases}$$

For $\ell$ odd and $\ell = 0$ the conclusion is clear, and for $\ell$ even and non-zero, we observe that the quadratic expression in $\ell$ above is minimized at $\ell = (k+1)/2$ (which is an integer, as $k$ is odd), and takes the value $(k^2 - 1)/4$, which is greater than $(k+1)/2$ for all odd $k \geq 5$ (though the two values are equal for $k = 3$).

Now observe that when this model is uprooted and subsequently rerooted at a new variable $i \in V$, we obtain a model on $k + 1$ variables, but with the variable $X_0$, introduced by uprooting, independent from the rest. Therefore, the model is effectively over only $k$ variables, and hence it follows that $\mathbb{L}_k(\nabla G \setminus \{i\})$ is tight for this rerooting, proving the claim. □

## 10 Additional Experimental Details and Results

In this section, we expand on the Experiments Section 7 of the main paper to provide:

- §10.1: Model structures and parameters used for libDAI
- §10.2: How we fit constants of the clamp selection heuristics
- §10.3: Additional experimental results
  - §10.3.1: Timings
  - §10.3.2: MAP inference
  - §10.3.3: Marginals
  - §10.3.4: Higher-order potentials over clusters of 5 and 6 variables
  - §10.3.5: Comparison of our heuristics to the maxtW heuristic used in [19]
  - §10.3.6: Larger models
- §10.4: Additional discussion

### 10.1 Model structures and parameters used for libDAI

In this section we give further information about the model structures used in our experiments, as well as the methods of approximate inference used. All potentials are pure $k$-potentials, as in §5, which for brevity we may write simply as a $k$-potential.

**Complete graphs** For complete graph experiments, there is a pure $k$-potential for each subset of $k$ variables, for $k = 1, 2, 3, 4$.

**Grids** All grids are square and toroidal. There is a 1-potential for each variable, and a 2-potential for each edge of the graph. There is a 3-potential for each possible "L-shaped" connected subgraph of size 3 (any of the four possible orientations), and a 4-potential for each cycle of size 4.

**Potentials** In our experiments, unless otherwise specified, the default is that all pure 2- and 4-potential coefficients are drawn independently from $\text{Unif}([-8, 8])$ distributions, while all pure 1- and 3-potential coefficients are set to 0. Using the notation of Section 7, in each experiment a parameter $W_{\max}$ is varied, and the default distribution of one class of pure potentials (either 1-, 2-, 3-, or 4-potentials) is overridden from the default specification to be replaced by coefficients from a $\text{Unif}([-W_{\max}, W_{\max}])$ distribution.

**LibDAI settings** In all cases, we use the junction tree algorithm with Hugin updates for exact inference. For approximate marginal inference, we use the LibDAI `HAK` implementation of [4], with precise parameters passed to MATLAB given by:

```
'[doubleloop=1,clusters=BETHE,init=UNIFORM,tol=1e-9,maxiter=10000]'.
```

For approximate marginal inference, we use the LibDAI BP loopy belief propagation implementation, with precise parameters passed to MATLAB given by:

```
'[inference=MAXPROD,updates=SEQFIX,logdomain=1,tol=1e-9,maxiter=10000,damping=0.0]'
```

## 10.2    How we fit constants of the clamp selection heuristics

In this section we give further details of how the K and G heuristics used in our experiments were fitted, expanding on the explanation given in Section 7. Using the notation developed in Section 7, the family of heuristics we consider maximize the following measure

$$\text{clamp-heuristic-measure}(i) = \sum_{i\in\mathcal{E}:|\mathcal{E}|=2} c_2 \tanh|t_2 a_\mathcal{E}| + \sum_{i\in\mathcal{E}:|\mathcal{E}|=4} c_4 \tanh|t_4 a_\mathcal{E}|, \qquad (11)$$

over $i \in V^+$, and are parametrized by the four scalars $t_2, c_2, t_4, c_4$. We first note that 11 is over-parametrized (since we are interested only in ranking the scores for each variable in $M^+$), so we take $c_2 = 1$. To fit the heuristic, we used gradient-free optimisation. For the K heuristic, we generated a collection of graphical models on $K_8$, and constructed a fitness function over the remaining parameters $t_2, c_4, t_4$, given by the average ranking of the rerooting selected by the heuristic for $\log Z$ estimation across our collection of complete graphs.

We then initialized the parameters $t_2 = 0, c_4 = 1, t_4 = 0$, and performed a local exploration of the parameter space dictated by a Gaussian random walk, updating our parameter settings when they led to an improvement in the value of the fitness function.

The G heuristic was constructed similarly, instead using a collection of grids to define the fitness function.

The precise values of the fitted heuristics are given below:

$$\text{K-heuristic-measure}(i) = \sum_{i\in\mathcal{E}:|\mathcal{E}|=2} c_2 \tanh|0.051 a_\mathcal{E}| + \sum_{i\in\mathcal{E}:|\mathcal{E}|=4} 0.091 \tanh|1.482 a_\mathcal{E}|,$$

$$\text{G-heuristic-measure}(i) = \sum_{i\in\mathcal{E}:|\mathcal{E}|=2} c_2 \tanh|0.019539 a_\mathcal{E}| + \sum_{i\in\mathcal{E}:|\mathcal{E}|=4} 0.3788 \tanh|0.033997 a_\mathcal{E}|.$$

The heuristic of [19], $maxtW$, applied only to pairwise models, and in the notation of our paper, was given by the following clamping score measure:

$$\text{clamp-heuristic-measure}(i) = \sum_{i\in\mathcal{E}:|\mathcal{E}|=2} \tanh|\frac{1}{2} a_\mathcal{E}|. \qquad (12)$$

Recognizing that the benefits of our heuristics appeared somewhat robust to exact parameter choice, when we extended analysis to 6-potentials in §10.3.4, we extended our K heuristic by eye (without fitting to any data, and before examining the results for higher order models), and explore a variant on the G heuristic. We used the following measures:

$$\text{K-heuristic-measure}(i) = \sum_{i\in\mathcal{E}:|\mathcal{E}|=2} \tanh|0.2 a_\mathcal{E}| + \sum_{i\in\mathcal{E}:|\mathcal{E}|=4} \frac{1}{3} \tanh|1.2 a_\mathcal{E}| + \sum_{i\in\mathcal{E}:|\mathcal{E}|=6} \frac{1}{5} \tanh|3 a_\mathcal{E}|,$$

$$\text{G-heuristic-measure}(i) = \sum_{i\in\mathcal{E}:|\mathcal{E}|=4} |a_\mathcal{E}|.$$

## 10.3    Additional experimental results

We provide the following:

- §10.3.1: Timings
- §10.3.2: MAP inference
- §10.3.3: Marginals

- §10.3.4: Higher-order potentials over clusters of 5 and 6 variables
- §10.3.5: Comparison of our heuristics to the maxtW heuristic used in [19]
- §10.3.6: Larger models

In all plots, if the red curve for the K heuristic is not visible, it coincides with the green curve for the G heuristic. We use consistent legends across all plots.

### 10.3.1 Timings

Times in seconds to run marginal inference (i.e. estimating $\log Z$ and marginals) using libDAI are shown in Figure 3. Inference for rerooted models took a similar amount of time as for the original model. We caution against relying heavily on the accuracy of these timings since we made no attempt to optimize our code for speed, and we ran our inference algorithms in a cluster environment beyond our control.

Time/sec to run marginal inference for $K_8$ complete hypergraphs (fully connected) on 8 variables.

vary $W_{\max}$ for 1-pots    vary $W_{\max}$ for 2-pots    vary $W_{\max}$ for 3-pots    vary $W_{\max}$ for 4-pots

Figure 3: Average time to perform marginal inference using libDAI over 20 runs. If not shown, $W_{\max}$ max coefficients for pure $k$-potentials are 0 for $k = 1$, 8 for $k = 2$, 0 for $k = 3$, 8 for $k = 4$. *Best* and *worst* refer to the rerootings which ex post gave the lowest error in estimating $\log Z$. See §10.3.1.

### 10.3.2 MAP inference

Results are shown in Figure 4. We observe here that rerooting does not help much when 1-pots are varied, but can provide great benefit for the other cases shown. The K heuristic (which was trained on complete graphs like the one we analyze here) performs well in all settings. Curiously, the G heuristic (which was trained only on grids) performs well when 2-pots or 4-pots are varied, but not when 3-pots are varied (though even here it does no worse than the original rooting). We aim to explore this further in future work.

Error in estimating MAP score for $K_8$ complete hypergraphs (fully connected) on 8 variables.

vary $W_{\max}$ for 1-pots    vary $W_{\max}$ for 2-pots    vary $W_{\max}$ for 3-pots    vary $W_{\max}$ for 4-pots

Figure 4: Average error in estimating MAP score using libDAI over 20 runs. If not shown, $W_{\max}$ max coefficients for pure $k$-potentials are 0 for $k = 1$, 8 for $k = 2$, 0 for $k = 3$, 8 for $k = 4$. *Best* and *worst* refer to the rerootings which ex post gave the lowest error in estimating $\log Z$. See §10.3.2.

### 10.3.3 Marginals

Results are shown in Figure 5. Our models were selected to present an interesting range of problems for partition function estimation, which led to marginals often being challenging to estimate. Still, results for marginal inference were often improved by rerooting.

We note that another natural way to estimate marginals is as the ratio of a clamped partition function to the original partition function. Since we have seen good evidence that rerooting can help significantly

with partition function estimation, it is reasonable to hope that in future work, we may observe significant benefits to marginal inference via this approach by using rerooting.

Error in estimating 1-marginals for $K_8$ complete hypergraphs (fully connected) on 8 variables.

vary $W_{max}$ for 1-pots     vary $W_{max}$ for 2-pots     vary $W_{max}$ for 3-pots     vary $W_{max}$ for 4-pots

Figure 5: Average $\ell_1$ error in estimating marginals (minimal representation corresponding to pure $k$-potentials, see §5) using libDAI over 20 runs. If not shown, $W_{max}$ max coefficients for pure $k$-potentials are 0 for $k = 1$, 8 for $k = 2$, 0 for $k = 3$, 8 for $k = 4$. *Best* and *worst* refer to the rerootings which ex post gave the lowest error in estimating $\log Z$. See §10.3.3.

### 10.3.4 Higher-order potentials over clusters of 5 and 6 variables

Results for a complete hypergraph $K_8$ on 8 variables, this time with potentials up to order 6, are shown in Figure 6. In all cases, rerooting using our heuristics is very helpful.

Error in estimating $\log Z$ (left) and MAP score (right) for $K_8$ hypergraphs on 8 variables with **potentials up to order 6**.

vary $W_{max}$ for 5-pots     vary $W_{max}$ for 6-pots     vary $W_{max}$ for 5-pots     vary $W_{max}$ for 6-pots

Figure 6: Average error in estimating $\log Z$ (left) and MAP score (right) using libDAI over 20 runs. If not shown, $W_{max}$ max coefficients for pure $k$-potentials are 0 for $k = 1$, 8 for $k = 2$, 0 for $k = 3$, 8 for $k = 4$. *Best* and *worst* refer to the rerootings which ex post gave the lowest error in estimating $\log Z$. See §10.3.4.

### 10.3.5 Comparison of our heuristics to the maxtW heuristic used in [19]

Results for a complete graph $K_8$ on 8 variables, this time with potentials only up to order 2, are shown in Figure 7. We have added the earlier maxtW heuristic used in [19], which using our notation corresponds to setting $t_2 = \frac{1}{2}$ in (5). Note that for the pairwise models considered here, the clamp heuristic constants for potentials of order higher than 2 are irrelevant.

We observe that our K and G heuristics (which were fit on different models with potentials up to order 4, so here are out of sample) achieve similar performance to the earlier maxtW heuristic, in fact yielding slightly better results. This is encouraging evidence for robustness of the simple form of heuristic score (5).

### 10.3.6 Larger models

Results for a complete hypergraph $K_{10}$ on 10 variables (potentials up to order 4) are shown in Figure 8. Results are qualitatively similar to those for smaller models in §7 of the main paper.

### 10.4 Additional discussion

When discussing pure $k$-potentials in §5, we observed that for a pure $k$-potential (which we showed must essentially be an $k$-potential) with $k$ an even number, $\theta_\mathcal{E}(x_\mathcal{E}) = \theta_\mathcal{E}(\overline{x}_\mathcal{E})$. This means that the coefficient of any such $k$-potential is invariant with respect to a flipping of all variables of the

Average abs(error) in $\log Z$ for $K_8$ complete **pairwise** graphs (fully connected) on 8 variables:
**adding earlier maxtW heuristic for comparison (our K and G heuristics coincide on these runs)**.

vary $W_{\max}$ for 1-pots

vary $W_{\max}$ for 2-pots

Figure 7: Error in estimating $\log Z$ for random pairwise models with various pure $k$-potentials over 20 runs. If not shown, $W_{\max}$ max coefficients for pure $k$-potentials are 8 for $k = 1$, and 8 for $k = 2$. K and G heuristics coincide. See §10.3.5.

Average abs(error) in $\log Z$ for $K_{10}$ complete hypergraphs (fully connected) on 10 variables.

vary $W_{\max}$ for 1-pots

vary $W_{\max}$ for 2-pots

vary $W_{\max}$ for 3-pots

vary $W_{\max}$ for 4-pots

Figure 8: Error in estimating $\log Z$ for random models with various pure $k$-potentials over 20 runs. If not shown, $W_{\max}$ max coefficients for pure $k$-potentials are 0 for $k = 1$, 8 for $k = 2$, 0 for $k = 3$, 8 for $k = 4$. See §10.3.6.

model (whereas the if $k$ is an odd number, the coefficient will flip sign). Hence for $k$ even, the sign of the coefficient may be regarded as a fundamental property of the potential.

When $k = 2$ this sign dicatates the submodularity or supermodularity of the 2-potential. If all potentials are pure 2-potentials with positive coefficients, then the model is *regular* or *ferromagnetic* and typically admits easier inference.

For higher $k$, this is no longer true. However, note that still if we represent a model's potentials in terms of pure $k$-potentials, and all have $k$ even with a positive coefficient, then the model is special in the sense that:

- The configurations of all 0s and all 1s must be mode configurations, typically with significantly higher probabilities than others.
- Inference will typically be relatively straightforward.
- If the model is rerooted, then this will effectively clamp all variables close to 0 or 1 and the error of approximate inference should be low.