[Reviews · NeurIPS 2017]

Reviewer 1



This paper presents a reparametrization method for inference in undirected graphical models. The method generalizes the uprooting and rerooting approach from binary pairwise graphical models to binary graphical models with factors of arbitrary arity. At the heart of the method is the observation that all non-unary potentials can be made symmetric, and once this has been done, the unary potentials can be changed to pairwise potentials to obtain a completely symmetric probability distribution, one where x and its complement have the exact same probability. This introduces one extra variable, making the inference problem harder. However, we can now "clamp" a different variable from the original distribution, and if we choose the right variable, approximate inference might perform better in the new model than in the original. Inference results in the reparametrized model easily translate to inference results in the original model. The idea is supported by a number of theorems. I believe the results, but I did not study the proofs in the supplementary material. In addition to the main result, which is that inference results in the new model can be mapped back to inference results in the original model, the authors also discuss how the reparametrized problems relate to each other in the marginal polytope and how factors can be decomposed into sums of "even" factors -- basically, parity over k variables. There are also some basic experiments on small, synthetic datasets which demonstrate that rerooting can yield better approximations in some cases. Given that the negligible cost of reparametrization and mapping the results, this seems like a promising way to improve inference. There's no guarantee that the results will be better, of course -- in general, rerooting could lead to better or worse results. Furthermore, we can clamp variables without rerooting, at the cost of doubling the inference time for each clamped variable. (Just run inference twice, once with each value clamped.) If rerooting is only done once, then you're getting at best a factor-of-two speedup compared to clamping. A factor of two is a small constant, and I didn't see any discussion of how rerooting could be extended to yield better results. So the practical benefit from the current work may be marginal, but the ideas are interesting and could be worth developing further in future work. Quality: I believe the quality is high, though as I mentioned, I have not studied the proofs in the supplementary material. The limitations are not discussed in detail, but this is typical for conference papers. Clarity: The writing was fairly clear. Figure 2 has very small text and is uninterpretable in a black and white printout. Originality: I believe this is original, but it's a natural evolution of previous work on rerooting. This is a meaningful increment, supported by theoretical results and token empirical results, but not transformative. Significance: As stated earlier, it's not clear that these methods will yield any improvements on real-world inference tasks (yet). And if they do, it's at best a factor-of-two speedup. Thus, this work will mainly be relevant to people working in the area of graphical models and probabilistic inference. It's still significant to this group, though, since the results are interesting and worthy of further exploration.

Reviewer 2



The authors present improvements over recent work on up/rerooting undirected graphical models; the novel step here is to consider potentials that involve more than two variables. The novelty seems to be the introduction of pure potentials, and the study of the consequences of using such pure potentials when dealing with uprooting/rerooting. The results show that up/rerooting increase the accuracy of approximate inference methods; this is a welcome result that is nicely studied empirically. Overall the paper brings an incremental but valuable contribution, and is very competent in its presentation. Small point: in Definition 1, what exactly happens if the edge consists only of a a pair of nodes? It becomes a single-node edge of some sort? A question: in Expression (3), perhaps it is 1[x_i NOT= x_j]? I do not see how the indicator can use the event [x_i = x_j] in this expression. A few suggestions concerning the text: - Introduction, line 3: I guess it should be "computing marginal probabilities", not "estimating marginal probabilities", right? - When there are several references together, please order them. - After Expression (3), do not start the sentence just with "(3) has", but perhaps "Expression (3) has". - I find that \mu_{V|U} is an awful symbol for marginal distributions; it looks like a conditional probability measure. Please use some other symbol (sometimes people use $V \downarrow U$ as a superscript to mean marginalisation). - Please remove "Proof in the Appedix" from the STATEMENT of Theorem 20.